# Decoupling Heterogeneous Features for Robust 3D Interacting Hand Poses Estimation

## ABSTRACT

Estimating the 3D poses of interacting hands from a monocular image is challenging due to the similarity in appearance between hand parts. Therefore, utilizing the appearance features alone tends to result in unreliable pose estimation. Existing approaches directly fuse the appearance features with position features, ignoring that the two types of features are heterogeneous. Here, the appearance features are derived from the RGB values of pixels, while the position features are mapped from the coordinates of pixels or joints. To address this problem, we present a novel framework called **D**ecoupled **F**eature **L**earning (**DFL**) for 3D pose estimation of interacting hands. By decoupling the appearance and position features, we facilitate the interactions within each feature type and those between both types of features. First, we compute the appearance relationships between the joint queries and the image feature maps; we utilize these relationships to aggregate each joint's appearance and position features. Second, we compute the 3D spatial relationships between hand joints using their position features; we utilize these relationships to guide the feature enhancement of joints. Third, we calculate appearance relationships and spatial relationships between the joints and image using the appearance and position features, respectively; we utilize these complementary relationships to promote the joints' location in the image. The two processes mentioned above are conducted iteratively. Finally, only the refined position features are used for hand pose estimation. This strategy avoids the step of mapping heterogeneous appearance features to hand-joint positions. Our method significantly outperforms state-of-the-art methods on the large-scale InterHand2.6M dataset. More impressively, our method exhibits strong generalization ability on in-the-wild images. The code will be released.

## CCS CONCEPTS

• **Computing methodologies** → **Computer vision**.

## KEYWORDS

3D Interacting Hand Poses Estimation, Feature Decoupling

## 1 INTRODUCTION

Estimating 3D poses of two interacting hands from a monocular image has great potential for applications in augmented reality (AR), virtual reality (VR), human-computer interaction, etc. Substantial

*ACM MM, 2024, Melbourne, Australia*
© 2024 Copyright held by the owner/author(s). Publication rights licensed to ACM.
ACM ISBN 978-x-xxxx-xxxx-x/YY/MM
https://doi.org/10.1145/nnnnnnn.nnnnnnn

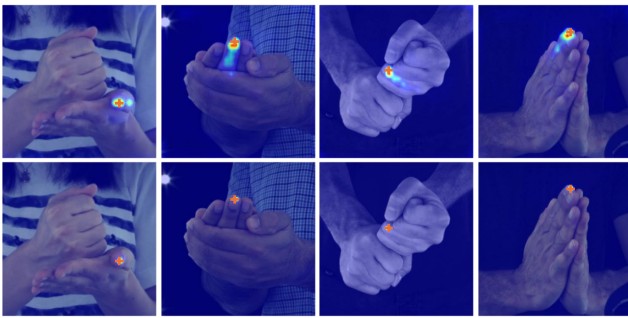

**Figure 1: Attention maps generated in the feature interaction between joints and the image are shown: the first row is from the baseline without the decoupling strategy. In the second row, attention maps from our method are depicted. The ground truth joint positions are marked with an orange cross. DFL precisely localizes the positions of the joints even in the presence of severe self-similarity.**

efforts have been dedicated to this field with the release of the large-scale interacting hand dataset [37]. Despite these achievements, it remains challenging due to the confusing appearance caused by self-similarity between hand parts.

In prior approaches, two main strategies have been employed to address the challenge of disambiguating similar appearance features. The first category of methods focuses on exploring the interaction between appearance features to extract more discriminative representations [9, 23, 33, 35, 53, 58]. The second category of solutions combines both position and appearance features to leverage both appearance and spatial information [13, 19, 28, 30, 42, 50, 54]. Nevertheless, the position features mapped from the coordinates of pixels or joints include spatial location and geometric structure information, while the appearance features mapped from the RGB values of pixels comprise color, texture, and other visual information. This distinction between these two feature types hinders the mutual facilitation process by direct interaction. Figure 1 demonstrates how the self-similarity in appearance can confuse the network (row 1) in accurately locating the joints in the image, emphasizing the challenges posed by the self-similarity of hand appearance.

To address the aforementioned issue, we present a novel framework called **D**ecoupled **F**eature **L**earning (**DFL**) for 3D interacting hand pose estimation. The main objective is effectively leveraging complementary heterogeneous features to mitigate the unreliable pose estimation caused by self-similarity between different hand parts. This is achieved by explicitly distinguishing between the appearance and position features and promoting the mutual interaction between different feature types and interaction within each feature type. As depicted in Figure 1, thanks to our decoupling

strategy, the network can accurately focus on the positions of the joints resulting in a sharp peak in the attention map (row 2).

Specifically, this paper utilizes a series of stacked modules. Firstly, it is observed that although each joint should not exhibit position preference in spatial across different images if the datasets are unbiased, they tend to show similar appearance patterns. Therefore, we employ queries to learn these appearance patterns. This approach differs from the design adopted in other studies [3, 13]. We compute the appearance relationship between joint queries and image feature maps. These relationships then guide aggregations of initial hand-joint position and appearance features from the feature maps. The initial features obtained are often unreliable due to appearance similarity.

To further improve the feature quality, the features are then iteratively enhanced through the feature interaction among the joints and between the joints and the image. The 3D spatial relationships embedded in joint positions are more robust to self-similarity than the relationships between appearances. So, in the first stage, we model the local and global spatial relationships between the joints' position features to guide the enhancement of joints. Subsequently, we leverage spatial and visual cues that are separately embedded in appearance and position features to achieve more accurate localization of joints in the image. Instead of directly fusing two types of features like the previous works [42, 50, 54], we extract and fuse their intra-relationships to promote the interaction between the features. These two complementary relationships guide the interactions within each feature type. Finally, the hand pose is regressed based on the refined position feature of the joints, rather than appearance features, which avoids the direct mapping between the appearance feature and hand poses [28, 30, 42, 50, 54]. It is worth mentioning that we adopt similarities-based computation [48] to extract intra-relationships.

Experimental results show that our method significantly outperforms existing state-of-the-art methods on the InterHand2.6M dataset. At the same time, we demonstrate that our method exhibits excellent generalization power when compared with state-of-the-art methods on multiple in-the-wild datasets.

## 2 RELATED WORK

### 2.1 Interacting Hand Pose Estimation.

The research on 3D hand pose estimation and shape reconstruction has a long history. The 3D hand pose estimation task aims to obtain joints' positions [5, 12, 56], while the 3D hand shape reconstruction task focuses on obtaining more dense representations such as mesh [6, 41, 55] or neural implicit surfaces [7, 27, 34]. Despite this distinction, the boundary between them is often unclear due to the potential for mutual transformation between mesh and joints through the parametric model [44] and inverse kinematics-based post-processing [29]. When introducing the parameterized model, the 3D interacting hand pose estimation approaches can be classified into model-free and model-based methods. The model-based approach is generally more robust due to the priors embedding in the model. However, model-free methods offer customizable topology and a relatively simpler calibration process for different individuals. This paper is model-free without relying on priors from the parametric model.

Pioneering works mainly estimate the single-hand pose based on depth [8, 18, 36, 49], RGB-D [2, 21, 39], or multi-view images [4, 11, 22]. Deep learning has enabled direct estimation from easily accessible monocular images [10, 20, 25]. Following decades of development, single-hand pose estimation has achieved significant success. As a result, the attention has shifted towards more challenging tasks, such as hand-object interaction pose estimation [15, 17, 32, 57], and interacting hands pose estimation.

Early methods in interacting hand pose estimation track articulated hands from observations by optimizing a series of defined energy functions [26, 40, 46]. These optimization-based methods converge slowly and are prone to get stuck in local minima. Therefore, hybrid methods integrate learning-based techniques to estimate intermediate visual representations for guiding the optimization process [1, 14, 38, 47]. However, these methods are not yet end-to-end. Besides, they typically cannot solely rely on a single image as input, leading to high costs and resource consumption. Thanks to large-scale dataset availability [37], the advancement in learning-based methods has mitigated these challenges.

Using monocular images as input exacerbates the issues of confusing image appearance caused by self-similarity. Rong et al. [45] attempted to regress the rough pose from ambiguous appearance features and further refined the initial pose by incorporating physical and geometric priors in the hand model. To make the extracted appearance features more distinctive for the joint regression, Meng [33] transformed overlapped interacting hand images to a single hand. Moon et al. [35] used a large-scale outdoor single-hand dataset with 2D annotations to enhance the backbone's feature extraction capability. The following work explores different intermediate representations to enhance appearance features. Kim et al. [23] used the joints' visibility to guide the heatmap enhancement. Fan et al. [9] proposed part segmentation to reduce the ambiguity of visual volume. Based on this, Yu et al. [53] further proposed various representations to disentangle two-hand features, such as the parameter map, hand center map, and cross-hand prior map. Finally, Zuo et al. [58] proposed the interaction adjacency heatmap which assigns denser visible features to those invisible joints. However, the enhanced appearance feature may not be reliable without incorporating prior information on skeletal structure. Following the idea that joints' appearance features and position features complement each other, Many methods fuse position into appearance features during feature interactions to implicitly leverage the relationship between appearance and inherent spatial information. The common practice is to obtain appearance features by projecting 3D positions. Zhang et al. [54] iteratively regressed poses based on mixed features. Wang et al. [50] utilized vertex-level fine-grained mixed features to promote mesh-image alignment. Ren et al. [42] completed occluded appearance features by projecting 3D joint features. Recently, Li et al. [28] learned noise distributions from fused features to refine mesh vertices and their projections. Apart from projection, Hampali et al. [13] proposed using the non-maximum suppression method to obtain potential 2D positions of joints and sample corresponding appearance features. Subsequently, the features of these joints interact with each other and are associated with joints to reduce self-similarity. Li et al. [30] directly used latent features from the backbone as appearance features and shared them between vertices. The interaction between the hands and the alignment between pose

and image is achieved through the proposed module. Jiang et al. [19] aggregated joints' appearance and position information using predefined anchor points.

Despite exploring the mutual assistance between appearance and positional features, the previous works are typically sub-optimal as the heterogeneous features hinder interaction between them.

## 2.2 Heterogeneous feature Disentanglement

In pose estimation tasks, the effective utilization of heterogeneous features of appearances and position features has not been extensively explored, unlike in tasks such as multi-modal and retrieval. Kim et al. [24] emphasized the challenge of mapping RGB values of pixels to heterogeneous joints' positions in human pose estimation and proposed using intermediate representations to mitigate this problem. Similarly, intermediate appearance representations are commonly employed in many methods for interacting hand pose estimation [9, 23, 33, 35, 53, 58]. Furthermore, previous methods often map the joints or pixel coordinates into position features, to enable the utilization of spatial relationships between them. However, they tend to overlook the heterogeneity between the two types of features [13, 19, 28, 30, 42, 50, 54]. In contrast, we decouple the heterogeneous appearance and position features and alleviate ambiguous appearances by mutual enhancement between the two feature types. In each module of our model, we employ different strategies to utilize the relationship from both types of features.

## 3 METHODS

This paper proposes a framework called **D**ecoupled **F**eature **L**earning (**DFL**) for 3D interacting hand pose estimation from a single RGB image. As shown in Figure 2, we adopt an encoder-decoder network structure. The encoder extracts multi-scale visual features with pixel-wise position features from the input image, see section 3.2. Then the decoder effectively utilizes complementary heterogeneous features to alleviate self-similarity and achieve accurate pose estimation, see section 3.3.

## 3.1 Preliminaries

In the task of 3D interacting hand poses estimation from a single RGB image, the objective is to predict the 3D positions of joints denoted as $P_{3D} \in \mathbb{R}^{2J \times 3}$ from the image $I \in \mathbb{R}^{H \times W \times 3}$ where $J$ is the number of joints in one hand. The joints' 3D representation can be derived by converting from 2.5D representation or parametric hand model. Our method adopts the 2.5D representation, which includes the coordinates in the image plane and the depth relative to the root joint.

## 3.2 Feature Extraction Encoder

Given an input image, the pre-trained ResNet50-FPN backbone is used to first extract multi-scale features $\{F_n \in \mathbb{R}^{H_n \times W_n \times C_F}\}_{n=0}^{N-1}$, where $H_n, W_n, C_F, N$ represent the height, width of the $n$-th feature map, the channel dimension, the number of feature scales respectively. To make the visual features more distinctive, we estimate probabilistic segmentation volumes $\{S_n \in \mathbb{R}^{H_n \times W_n \times C_S}\}_{n=0}^{N-1}$ from the last feature map to represent identity information for each pixel, inspired by [9]. The low-resolution segmentation map is obtained by down-sampling the high-resolution segmentation map. Each

volume channel represents the probability of one of the $C_S$ classes, where $C_S = 33$, including 16 hand part classes for each hand and 1 background class. Following that, multi-scale features and segmentation volumes are concatenated along the channel dimension to form the image appearance features $\{F_n^a \in \mathbb{R}^{H_n \times W_n \times (C_F + C_S)}\}_{n=0}^{N-1}$. Finally, we apply position encoding to appearance features and map the obtained position embedding to form the image position features $\{F_n^p \in \mathbb{R}^{H_n \times W_n \times C_P}\}_{n=0}^{N-1}$. The obtained multi-scale features supply the decoder with information at varying granularity and notably reduce the computational burden compared to using only high-resolution feature maps.

Although the intermediate representation mentioned above provides rich visual cues to alleviate appearance ambiguity. However, the visual features are only enhanced through intermediate representations which are estimated by the interaction between appearance features, such features remain unreliable for regressing accurate joint positions due to appearance self-similarity. Considering the complementarity but heterogeneity between appearance features and position features, exploring appropriate methods for effectively leveraging them is crucial.

## 3.3 Decoupling Heterogeneous Feature Decoder

In the previous method, the two types of features were directly fused and then enhanced through feature interactions. The fused features were subsequently mapped to regress poses. Due to the inherent differences between the two types of features, such interaction and mapping processes are challenging.

Based on the above observation, we propose explicitly decoupling the two heterogeneous features. Simultaneously, our method allows for independent modeling of each feature type, taking into account their respective characteristics. Moreover, it preserves the exchange of complementary information between the two types of features. Following this design principle, we employ different strategies to effectively utilize the relationships from both appearance and position features in each module of our model. Specifically, we first extract the joints' initial appearance and position features with the guidance of appearance relationships from the image appearance and position features. Next, we iteratively update the joints' appearance and position features with the guidance of the spatial relationships between joints and complementary relationships of spatial and appearance between joints and the image. The total number of iterations is $T = N - 1$.

*3.3.1 Constructing Initial Joints' Features.* In light of the absence of a consistent spatial pattern but the presence of similar appearance patterns of each joint across different images, we compute the appearance relationship between learnable queries $Q^a \in \mathbb{R}^{2J \times (C_F + C_S)}$ and the appearance features of the image without the position features. The relationships extraction process can be formalized as follows:

$$A_0^a = Softmax\left(\frac{DP(Q^a, F_0^a)}{\sqrt{C_F + C_S}}\right). \tag{1}$$

where $DP(M_1, M_2)$ denotes Dot Product computation representing the pairwise dot product operation between the row vectors of matrix $M_1$ and the column vectors of matrix $M_2$. $A^a \in \mathbb{R}^{2J \times (H_0 W_0)}$ denotes the appearance relationship matrix.

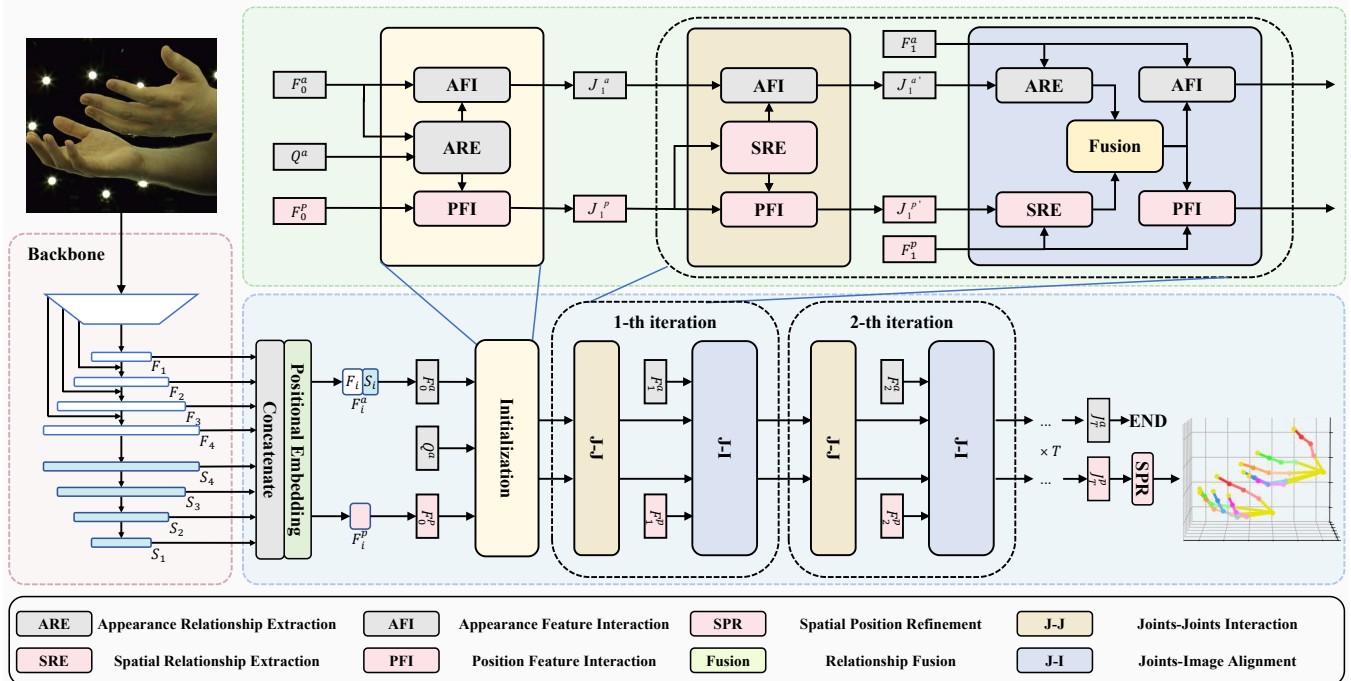

**Figure 2: Illustration of our decoupling heterogeneous feature framework: the multi-scale virtual features extracted from the backbone. Thanks to the decoupling strategy, initial joints' features are obtained using appearance relationship embedding in appearance features (Initialization). Then, the features are enhanced by utilizing the 3D spatial relationships embedded in position features (J-J). Subsequently, the complementary position and appearance relationships jointly promote the joints' location in the image (J-I). Finally, position features are refined (SPR) and used for the regression of poses.**

Therefore, the joint queries were only to learn the appearance patterns of joints. The appearance relationships are then used to guide the aggregation of each joint's initial appearance and position features from the image appearance and position feature as

$$\mathbf{J}_1^a = \mathbf{A}_0^a \mathbf{F}_t^a, \mathbf{J}_1^p = \mathbf{A}_0^a \mathbf{F}_t^p. \quad (2)$$

*3.3.2 Joints-Joints Spatial Relationship Modeling.* Compared to appearance relationships, spatial relationships embedded in position features are more robust in dealing with situations of severe self-similarity. Therefore, we model the 3D spatial relationships $\mathbf{A}_t^{p'} \in \mathbb{R}^{2J \times 2J}$ embedded in the position features, similar to 2. These relationships guide the enhancement of the features, resulting in improved position and appearance features $\mathbf{J}_t^{a'} \in \mathbb{R}^{2J \times (C_F + C_S)}$ and $\mathbf{J}_t^{p'} \in \mathbb{R}^{2J \times C_P}$, respectively. This enhancement process follows a similar formulation as described in Equation 2 where the subscript $t$ represents $t$-th iteration.

We further utilize joints' position feature in the final iterations to model local and global skeletal structure priors like [52]. It incorporates lightweight GCN and an attention module called the spatial position refinement module.

*3.3.3 Joints-Image Position-Appearance Relationship Modeling.* To better facilitate the localization of joints in the image, we leverage spatial and visual cues that are separately embedded in appearance and position features. Instead of directly fusing two types of features

like the previous works [3, 13], we extract spatial relationships $\mathbf{A}_t^p \in \mathbb{R}^{2J \times (H_t W_t)}$ and appearance relationships $\mathbf{A}_t^a \in \mathbb{R}^{2J \times (H_t W_t)}$, respectively. Then, we add the two types of relationships together to fuse them. These two complementary relationships guide the interactions within each feature type.

The process outlined above can be implemented as (please refer to the Appendix for details):

$$\mathbf{Q}_t = Concat(\mathbf{J}_t^{a'}, \mathbf{J}_t^{p'}), \mathbf{K}_t = \mathbf{V}_t = Concat(\mathbf{F}_t^a, \mathbf{F}_t^p),$$
$$\mathbf{J}_{t+1}^a, \mathbf{J}_{t+1}^p = Split(Attn(\mathbf{Q}_t, \mathbf{K}_t, \mathbf{V}_t)). \quad (3)$$

Finally, the enhanced position features are used for regressing the pose $\mathbf{P}_{2.5D} \in \mathbb{R}^{2J \times 3}$ by a linear layer.

*3.3.4 Further Discussion.* Our method extracts relationships within each feature type as proxies for the interaction between the features. Different relationships are utilized in the three modules mentioned above:

$$\mathbf{A}_{\mathcal{M}1} = \mathbf{Q}_a^T \mathbf{K}_a,$$
$$\mathbf{A}_{\mathcal{M}2} = \mathbf{Q}_p^T \mathbf{K}_p,$$
$$\mathbf{A}_{\mathcal{M}3} = \mathbf{Q}_a^T \mathbf{K}_a + \mathbf{Q}_p^T \mathbf{K}_p. \quad (4)$$

where $\mathbf{Q}$ and $\mathbf{K}$ represent the feature matrices. $\mathcal{M}n$ represents the $n$-th module. Subscripts $a$ and $p$ respectively denote the appearance and position features. By exchanging relationships within the

features, mutual promotion between features is facilitated. Then these relationships guide the interaction within the features. This is quite different from the previous work, as they involve direct interactions or mutual mappings between heterogeneous features to implement the mutual promotion, such as:

$$\begin{aligned} \mathbf{C}_{fusion} &= \mathbf{Q}_a \oplus \mathbf{K}_p; \\ \mathbf{C}_p &= \mathcal{F}_{a->p}(\mathbf{Q}_a). \end{aligned} \quad (5)$$

where $\mathbf{C}_n$ represents the output of the n-th class operation. The symbol $\oplus$ represents various computational operations such as matrix addition, matrix multiplication, etc. These operations are utilized to fuse two features. $\mathcal{F}_{a->p}$ denotes the function that maps appearance features to position features.

## 3.4 Loss Functions

The loss function can divided into two groups, including the joint loss and pixel-wise loss.

*3.4.1 Joints Loss.* Following [19], we use the combination of two $smooth_{\mathcal{L}_1}$ losses to supervise the final predicted joints as:

$$\mathcal{L}_{\mathbf{2.5D}} = \alpha \mathcal{L}_{\tau_1}(\hat{\mathbf{P}}_{uv} - \mathbf{P}_{uv}) + \beta \mathcal{L}_{\tau_2}(\hat{\mathbf{P}}_d - \mathbf{P}_d). \quad (6)$$

where $\mathbf{P}_{uv}$ and $\mathbf{P}_d$ denotes in-plane and depth coordinate of joints, respectively. The parameter $\alpha$ is set to 0.5, and the parameter $\beta$ is set to 1. Besides, $\hat{}$ denotes the predicted values. Assuming $\mathbf{X} \in \mathbb{R}^{m \times n}$, the term $\mathcal{L}_{\tau}$ is defined as [43]:

$$\mathcal{L}_{\tau}(\mathbf{X}) = \begin{cases} \frac{1}{2\tau} \sum_{i=1}^{m} \sum_{j=1}^{n} x_{ij}^2, & \text{for } |x_{ij}| < \tau, \\ \sum_{i=1}^{m} \sum_{j=1}^{n} |x_{ij}| - \frac{\tau}{2}, & \text{otherwise.} \end{cases} \quad (7)$$

where $\tau_1$, $\tau_2$ are set to 1, 3 for better smoothing the depth value.

We also employ $\mathcal{L}_{\mathbf{2D}}$ to supervise intermediate results at each iteration. Here, $\alpha$, $\beta$ and $\tau_1$ are set to 1, 0, 1, respectively.

*3.4.2 Pixel-wise Loss.* We employ the multi-class focal loss [31] and soft dice loss to supervise part segmentation to reduce feature ambiguity. The multi-class focal loss is defined as:

$$\mathcal{L}_{Focal} = - \sum_{m=1}^{H_{N-1}} \sum_{n=1}^{W_{N-1}} \sum_{j=1}^{C_S} \mathbf{T}_{mn;j} (1 - \sigma(\mathbf{S}_{mn;j})^\gamma) \log(\sigma(\mathbf{S}_{mn;j})). \quad (8)$$

where $\mathbf{T}_{mn} \in \mathbf{R}^{C_S}$ is one-hot vector and $\mathbf{T}_{mn;j} = 1$ when $j$ is true label. $\sigma(\cdot)$ is a softmax operation. The parameter $\gamma$ makes the model focus more on challenging samples. If $\gamma = 0$, focal loss equals cross-entropy loss. $\gamma$ is set to 2.

The multi-class soft dice loss is defined as:

$$\mathcal{L}_{Dice} = 1 - \frac{1}{C_S} \sum_{j=1}^{C_S} \frac{\sum_{m=1}^{H_{N-1}} \sum_{n=1}^{W_{N-1}} 2\mathbf{T}_{mn;j} \sigma(\mathbf{S}_{mn;j}) + \epsilon}{\sum_{m=1}^{H_{N-1}} \sum_{n=1}^{W_{N-1}} \mathbf{T}_{mn;j} + \sigma(\mathbf{S}_{mn;j}) + \epsilon}. \quad (9)$$

The $\epsilon$ is a smoothing coefficient that ensures numerical stability and can also smooth the loss.

The final segmentation loss is:

$$\mathcal{L}_{seg} = \mathcal{L}_{Focal} + \mathcal{L}_{Dice}. \quad (10)$$

*3.4.3 Total Loss.* The total loss is the weighted sum of the individual losses mentioned above and can formulated as:

$$\mathcal{L}_{total} = \mathcal{L}_{2D} + \lambda_1 \mathcal{L}_{2.5D} + \lambda_2 \mathcal{L}_{seg}. \quad (11)$$

where $\lambda_1$ and $\lambda_2$ are set to 3 and 1 to balance losses.

## 4 EXPERIMENTS

### 4.1 Datasets and Metrics

*4.1.1 Datasets.* We primarily evaluate our method on the Interhand2.6M. Interhand2.6M is the only published large-scale dataset for monocular interactive hand pose estimation tasks, which include complex interactive hand pose. It contains 1.36M training data and 849K testing data, including single-hand (SH) and interacting hand (IH) images. For a fair comparison, we train our model and report the results on 5 FPS SH+IH subsets with H+M annotations following the model-free common practice [19, 37] compared to model-free methods. When comparing with model-based methods, we train and test our model in a filtered dataset following their setting [30, 42].

Because the Interhand2.6M dataset has minimal background variations, we evaluate the model's generalization capability in the HIC dataset from Tzionas et al. [16]. To the best of our knowledge, this is the only publicly available RGB dataset that provides 3D joint annotations for hands engaged in strong interactions under natural lighting conditions. Following [35], 732 images were used. We also conducted qualitative experiments on the RGB2Hands [51] dataset.

*4.1.2 Metrics.* Firstly, the Mean Per Joint Position Error (MPJPE) is adopted for evaluation. It is defined as the Euclidean distance between the predicted and ground truth 3D positions after aligning two hands with their respective root joints. Following common practices in model-free methods, we use the wrist joint as the root joint and do not scale the estimated pose using the gt bone lengths when computing MPJPE. Second, we report the Percentage of Correct Keypoints (PCK) and Area Under the Curve (AUC) between 0 and 50 millimeters. Besides, FPS is used to evaluate the inference speed. All methods are tested on a single TitanV GPU.

### 4.2 Implementation Details

All implementations are based on PyTorch. The Adam optimizer with an initial learning rate of 1e-4 is used to train our network. The model was trained for 50 epochs with a batch size 64 using four NVIDIA Titan V GPUs. The learning rate decayed at the 24th and 35th epochs. We perform data augmentation, including random horizontal flipping, random rotation, random scaling, and random translation. Following [19, 42], we crop out the region of the hand based on their bounding box and resize it to 256×256.

### 4.3 Ablation Study

We conduct ablation experiments on the interhand2.6m dataset. In the following experiments, the number of iterations is set to 3, unless otherwise specified.

*4.3.1 Baseline.* We first explore several network variations that can serve as baselines. In these baseline variations, we do not employ the proposed feature decoupling strategy in the decoder. As

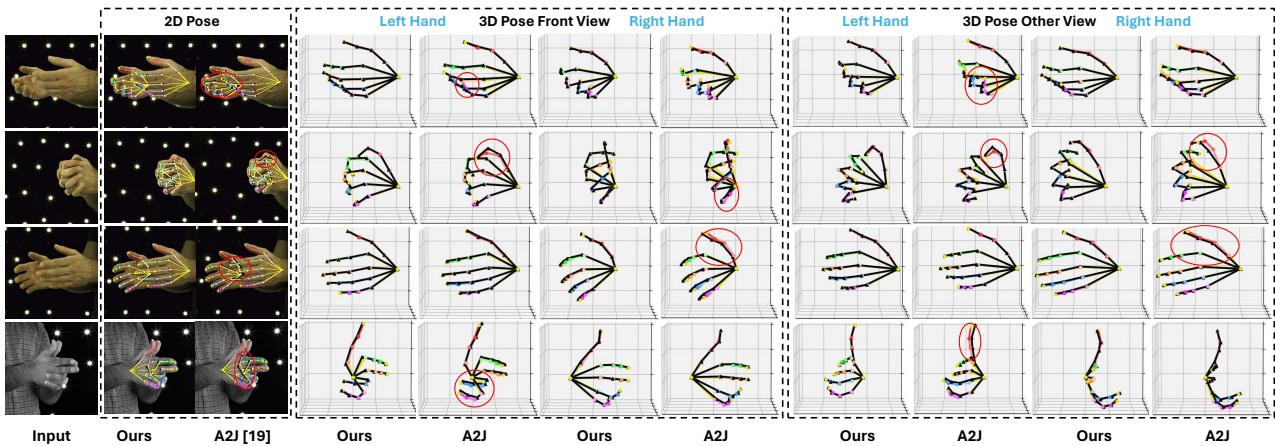

**Figure 3: Qualitative results of A2J [19] and ours on InterHand2.6M dataset. The ground truth of 2D and 3D poses are represented in black color. For better visualization, we present the left and right hands separately and align the root nodes of both hands. Besides, the lighting is adjusted for better display (not model input).**

**Table 1: Ablation study on baseline. J-I and J-J denote Joints-Image interaction and Joints-Joints interaction, respectively. SPR means Spatial Position Refinement module.**

| ID | J-I | J-J | SPR | MPJPE (mm)↓ | | |
|---|---|---|---|---|---|---|
| | | | | Single | Two | All |
| 1 | | | | 9.25 | 14.27 | 11.94 |
| 2 | √ | | | 8.30 | 11.78 | 10.16 |
| 3 | √ | √ | | 8.05 | 11.18 | 9.72 |
| 4 | √ | √ | √ | 7.98 | 11.05 | 9.62 |

shown in Table 1, the basic method (ID 1), which directly estimates pose from the average pooled features extracted by the encoder, performs poorly. Subsequently, we introduce learnable queries and iteratively perform feature interaction between joints and the multiscale virtual features using cross-attention (ID 2). Since the spatial relationships between joints are crucial for alleviating the self-similarity issue, we further add the joints-joints interaction module, thus forming the two-stage pipeline (ID 3). The empirical results demonstrate that both stages enhance the network's performance; therefore, we adopt them in all subsequent experiments by default. To further capture both local and global spatial relationships, we employ a spatial position refinement module (ID 4).

*4.3.2 Decoupling Strategy in Different Module.* In this section, we experiment with different decoupling strategies in each module. The first row in Table 3 is our final model with the best feature decoupling strategy in each module. Compared to the model without any feature decoupling strategy in the last row of Table 1, DFL significantly improves by 0.63mm, 1.23mm, and 0.95mm. Subsequent experiments will employ DFL as the baseline method.

The second row of Table 3 demonstrates the impact of learning different joint patterns using learnable queries to guide the construction of initial joints' features from the image. Here, the query interacts with different features to learn the corresponding pattern. The results suggest that learning joint position patterns hinders

**Table 2: Ablation study on decoupling strategy in different stages.**

| Stage | A | P | MPJPE (mm)↓ | | |
|---|---|---|---|---|---|
| | | | Single | Two | All |
| Best Model | - | - | 7.35 | 9.82 | 8.67 |
| Initialization | | √ | 7.36 | 10.19 | 8.93 |
| | √ | √ | 7.57 | 10.05 | 8.89 |
| J-I Iteration | √ | | 7.68 | 10.19 | 9.02 |
| | | √ | 7.59 | 10.20 | 8.98 |
| J-J Iteration | √ | | 7.60 | 10.22 | 9.00 |
| | √ | √ | 7.49 | 10.11 | 8.89 |
| Regression | √ | | 7.76 | 9.99 | 8.94 |
| | √ | √ | 7.45 | 9.89 | 8.75 |

**Table 3: Ablation study on the number of iterations.**

| Count | MPJPE (mm)↓ | | |
|---|---|---|---|
| | Single | Two | All |
| 2 | 7.56 | 9.86 | 8.78 |
| 3 | 7.35 | 9.82 | 8.67 |
| 4 | 7.36 | 9.78 | 8.65 |
| 5 | 7.30 | 9.78 | 8.62 |

network performance. This observation may be attributed to the prior positions of the joints do not exhibit spatial preference.

The third row of Table 3 presents the impact of using different relationships in joint-image interaction processes. It shows that both spatial and appearance relationships contribute to the accurate localization of joints in the image.

The fourth row of Table 3 shows the performance of using different relationships during joint-joints interaction processes. We observed a slight decrease in performance when introducing appearance relationships. These results suggest that appearance relationships are ineffective in enhancing features due to self-similarity.

**Table 4: Comparison with state-of-the-art model-based and model-free methods on InterHand2.6M. MPJPE, FPS, and model size are reported.†denotes the result of the model in the filtered IH dataset following the model-based method.**

| Methods | MPJPE(mm)↓ | | | FPS↑ (s) | Model ↓ Size(M) |
|---|---|---|---|---|---|
| | Single | Two | All | | |
| **Model-based** | | | | | |
| IntagHand [30] † | - | 15.74 | - | 19.46 | **39** |
| DIR [42] † | - | 12.69 | - | 13.67 | 55 |
| **Model-free** | | | | | |
| InterHand [37] | 12.16 | 16.02 | 14.22 | **58.26** | 47 |
| DIGIT [9] | 11.32 | 15.57 | - | 15.36 | 41 |
| KPT [13] | 10.99 | 14.34 | 12.78 | 25.57 | 48 |
| A2J [13] | 8.10 | 10.96 | 9.63 | 19.21 | 42 |
| Ours | **7.35** | **9.82** | **8.67** | 20.01 | 42 |
| A2J [13]† | - | 11.90 | - | 19.21 | 42 |
| Ours† | - | **10.68** | - | 20.01 | 42 |

**Table 5: Comparison with state-of-the-art model-free methods on HIC. MPJPE is reported.**

| Methods | MPJPE (mm)↓ |
|---|---|
| Interhand [37] | 29.75 |
| DIGIT [9] | 20.98 |
| KPT [13] | 26.38 |
| A2J [13] | 23.51 |
| Ours | **20.71** |

The last row of Table 3 presents the impact of employing different features to regress pose. The findings indicate the successful decoupling of the two feature types, with decoupled appearance features exhibiting no discernible positive impact on the pose regression.

*4.3.3 Ablation Study On the Number of Iterations.* Table 3 demonstrates that there is limited performance improvement when increasing the number of iterations beyond three times. Considering that more iterations lead to larger model sizes and higher computational costs, the network iterates three times in total to strike a balance between performance and efficiency.

## 4.4 Comparisons to State-of-the-arts Methods

*4.4.1 Comparisons on Interhand2.6M.* We first compare our method with the most relevant model-free methods. We follow the official data split to train and test our model. Table 4 shows that DFL significantly outperforms the state-of-the-art model-free 3D interacting hand pose estimation method [19] under all scenarios. Specifically, compared to the SOTA model-free methods, the improvement of DFL is 0.75mm, 1.14mm, and 0.96mm respectively. In addition, we have comparable FPS and model sizes compared to SOTA methods. When comparing the model-based approaches, we retrain and retest both DFL and [19] following their dataset setting to ensure a fair comparison. Furthermore, since the predicted bone length is

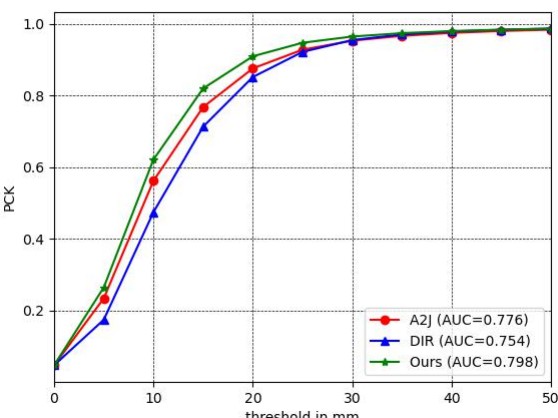

**Figure 4: Comparison with SOTA model-free and model-based methods on InterHand2.6M dataset.**

meaningful for the pose estimation task, we follow the model-free practice and do not use ground truth bone length information during evaluation. We get the result by running their released code and checkpoints. Results show that our results significantly surpass SOTA model-based methods [42] by 2.01mm while having faster inference speed and a smaller model size.

In addition, we compared our method with SOTA model-free and model-based approaches using both PCK and AUC metrics. Figure 4 shows that our method outperforms them at almost all error thresholds and achieves the highest AUC score.

*4.4.2 Comparisons on HIC.* To evaluate the generalization of our method, we test it on in-the-wild images. Table 5 shows the superiority of our method compared to other approaches. Although DFL was not specifically designed for generalization, it achieves state-of-the-art results. Notably, [9] demonstrates strong generalization ability compared to previous model-free methods. But we still outperform it by 0.27mm, demonstrating the robustness of our approach.

## 4.5 Qualitative Results

*4.5.1 Qualitative results on Interhand2.6M.* We present the qualitative results of our method on the Interhand2.6M in Figure 3. Compared to [9], our method significantly reduces ambiguity caused by self-similarity hand appearance. On one hand, the spatial relationships between joints promote a more reasonable spatial hand pose configuration. On the other hand, the complementary position and appearance relationship between joints and image promote better joint-image alignment. Even under challenging poses, our method achieves accurate pose estimation (row 2, row 4).

*4.5.2 Qualitative results on in-the-wild image.* Similar to [9], we also qualitatively tested the generalization ability of our method on in-the-wild images. It is worth mentioning we only trained our model on the interhand2.6M dataset without fine-tuning it on any other datasets. As shown in Figure 5, our method demonstrates good generalization under various lighting conditions and backgrounds.

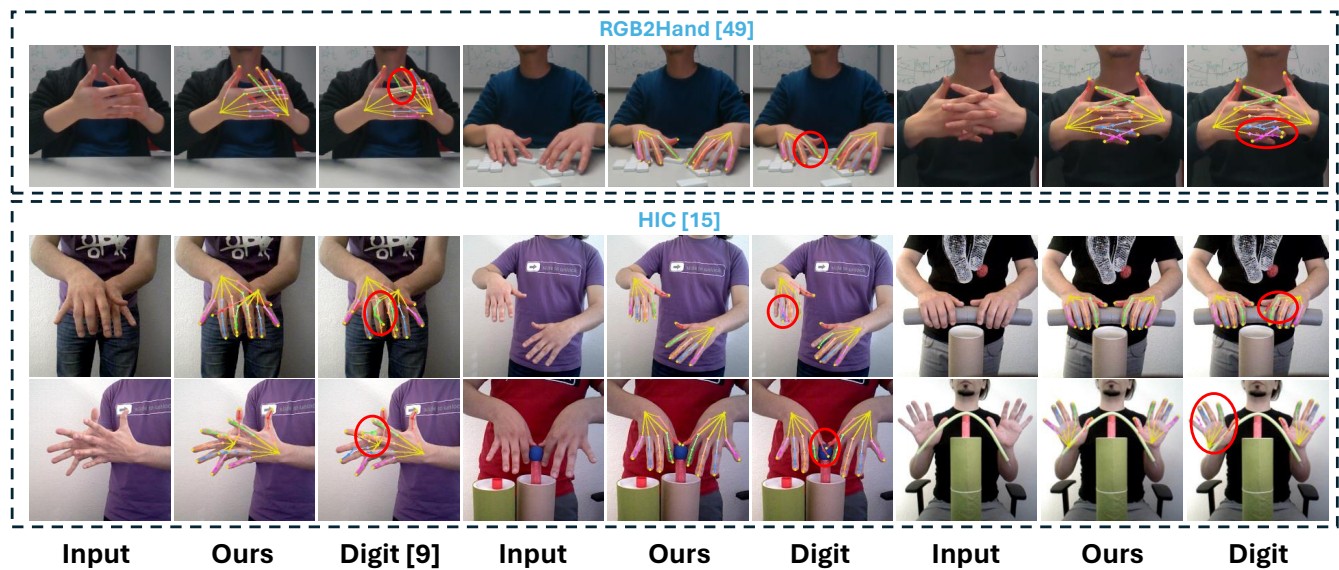

**Figure 5: Qualitative results of Digit [9] and ours on the in-the-wild images. The first row corresponds to the RGB2Hands dataset, while the results from the second and third row corresponds to the HIC dataset.**

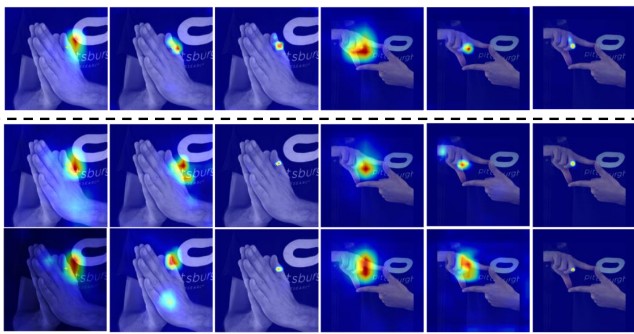

**Figure 6: The first row displays the 3-iteration attention map from the baseline without the decoupling strategy. The second and last rows respectively demonstrate the 3-iteration attention maps of the appearance features and position features.**

Due to the introduction of spatial relationship modeling between joints, our method is less likely to generate unreasonable poses although we do not explicitly use physical constraints. Furthermore, our method demonstrates robustness against self-similarity in appearance in cases where both hands are in close interaction compared to the previous SOTA method [9].

*4.5.3 Qualitative Analysis.* We investigate how the decoupled appearance and position features work together to reduce appearance ambiguity in interacting hand pose estimation. Figure 6 shows attention maps generated from the feature interaction between joints and the image. The first row displays the 3-iteration attention map from the baseline without the decoupling strategy. Due to the heterogeneity between position features and appearance, it is

challenging to achieve mutual enhancement between them. Therefore, when there is severe self-similarity in appearance patterns, it is hard to accurately focus on the location of joints. With the proposed decoupling strategy, the position features utilize spatial cues, while the appearance features employ visual cues to jointly promote the joints' location in the image. The second and last row in Figure 6 respectively demonstrate the 3-iteration attention maps of the appearance features and position features. The two types of features mutually enhance each other, resulting in both appearance and position features being able to independently and accurately localize the positions of the joints in the last iteration (Please refer to the supplementary materials for more visualizations).

## 5 CONCLUSION

This paper proposes the DFL framework to effectively leverage complementary heterogeneous features to mitigate self-similarity between different hand parts. In DFL, we explicitly decouple appearance and position features, which facilitate the interactions within each feature type and those between both types of features. Thanks to such a decoupling strategy, initial features are first obtained with the guidance of appearance relationships. Next, the features are enhanced by the guidance of spatial relationships. Then, complementary appearance and position relationships are fused to promote the location of joints in the image. Finally, only positional features are used to regress the pose. The experiments conducted on InterHand2.6M indicate that our method significantly outperforms the previous state-of-the-art approach. Moreover, the evaluation of images captured in the wild scenarios highlights the robust generalization ability of our method.

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
