# OpenReview forum: "Decoupling Heterogeneous Features for Robust 3D Interacting Hand Poses Estimation"
_acmmm.org/ACMMM/2024/Conference — MM2024 Poster_

### Official Review · Reviewer_FqoD · 2024-05-08

**Rating:** 4
**Confidence:** 4

**Summary:**

The paper introduces a novel framework called Decoupled Feature Learning (DFL) for estimating the 3D poses of two interacting hands from a single RGB image. The authors address the challenge of self-similarity in hand appearances by decoupling appearance and position features, allowing for more effective interactions within and between these feature types. The framework computes relationships between joint queries and image feature maps, spatial relationships between hand joints, and appearance and spatial relationships between joints and the image. The method iteratively refines these features and finally uses only the refined position features for hand pose estimation. The authors claim significant performance improvements over state-of-the-art methods on the InterHand2.6M dataset and strong generalization on in-the-wild images.

**Strengths:**

- The paper presents a novel approach to the problem of 3D hand pose estimation by explicitly decoupling appearance and position features, which is a unique contribution to the field.
- The theoretical foundation for decoupling features is well-explained, and the proposed DFL framework is logically sound, with a clear methodology for handling heterogeneous features.
- The paper provides a thorough evaluation on the InterHand2.6M dataset and demonstrates generalization capabilities on in-the-wild images, which is commendable.

**Limitations:**

- The paper does not discuss the computational complexity of the proposed method. While the method may be effective, it could be computationally intensive due to the iterative nature of the process, which might limit its applicability in real-time systems.
- While the method shows good performance, the paper could benefit from a discussion on the explainability of the model, i.e., how the decoupling of features leads to the improved localization of hand joints.

**Suitability:**

2

---

### Official Review · Reviewer_jBhm · 2024-05-09

**Rating:** 2
**Confidence:** 3

**Summary:**

This paper introduces a framework called Decoupled Feature Learning (DFL) for estimating 3D poses of interacting hands from monocular images. DFL promotes interactions within each feature type and between the two feature types by decoupling appearance and position features. The method first calculates the appearance relationship between joint queries and image feature maps, and uses these relationships to aggregate the appearance and position features of each joint. Then, it uses position features to calculate the 3D spatial relationship between hand joints to guide the feature enhancement of joints. Next, it uses appearance and position features to calculate the appearance and spatial relationships between joints and images, respectively, and uses these complementary relationships to promote the positioning of joints in images. These two processes are iteratively performed. Finally, only refined position features are used for hand pose estimation, avoiding the step of mapping heterogeneous appearance features to hand joint positions. It achieves high accuracy on the large InterHand2.6M dataset.

**Strengths:**

1.The proposed method achieves high accuracy on the InterHand2.6M dataset.
2.The third section of this paper provides a very detailed description of the method, enabling readers to have a clearer understanding of the structural details of the model.

**Limitations:**

1. The concept described in this paper is doubtful because the position of the joint itself is obtained based on the apperence features of the hand, so the so-called "Decoupling" motivation of position and apperence  is not understandable. Although the modules proposed in the experiment contribute to the results, the modules proposed in this paper can also be interpreted in other ways, such as using more image features to optimize the position. In summary, the "story" described in this paper is not convincing. The article uses Figure 1 as a basis for supporting the proposed concept, but does not explain in detail the specific meaning of Figure 1 and how it was obtained.
2. There are many typographical and textual errors in the article, such as Table 3 in Section 4.3.2 should be Table 2, etc
3. The description of the experiment is worth further refining and optimizing. For example, in Table 3, the single indicator when Count is 3 is better than when Count is 4, but the authors did not provide an explanation for this. More recent methods should be introduced in Section 4.4 to compare with this work.
4. Why is this work limited to estimating interacting hand poses? It seems that it can also be used for estimating single-hand poses. So the definition of this paper's work needs to be discussed. *I would like to know how this work performs on the single-hand pose estimation datasets HO3Dv2 and FrieHAND.*

**Suitability:**

2

---

### Official Review · Reviewer_8rMr · 2024-05-20

**Rating:** 4
**Confidence:** 2

**Summary:**

This paper proposes a framework called Decoupled Feature Learning (DFL) for estimating 3D poses of interacting hands from monocular images. The traditional approach of fusing appearance and position features directly is ineffective due to their heterogeneity. By decoupling these features, DFL facilitates intra-type and cross-type feature interactions. DFL outperforms state-of-the-art methods on the InterHand2.6M dataset and exhibits generalization on real-world images.

**Strengths:**

[Novelty/relevance] Proposed approach is novel and builds well on the existing literature of appearance and position features learning based methods for 3D interacting hand poses estimation.

[Clarity/experiments] Paper is very well written with detailed experiments as well as ablation study.

[Performance] Superior performance is reported on public benchmarks (InterHand2.6M and HIC).

**Limitations:**

1. Technical concerns

1.1  Why is it that the experimental results presented in this paper indicate that the model-free method outperforms the model-based method in terms of accuracy, whereas this finding contrasts with the conclusions drawn from the results reported in the A2J paper?

Jiang, Changlong, et al. "A2J-Transformer: Anchor-to-Joint Transformer Network for 3D Interacting Hand Pose Estimation from a Single RGB Image." Proceedings of the IEEE/CVF Conference on Computer Vision and Pattern Recognition. 2023.

1.2 If the main focus of this paper is on the model-free approach, how was the implementation of the model-based results specifically designed?


2. Minor points

2.1 “The first row in Table 3 is our final model…” (line 628) May be is Table 2.

2.2 “KPT [13], A2J [13]” (line 712)

2.3 Some references are duplicated (7-8).

2.4 Reference citation format is not uniform.

**Suitability:**

3

---

### Official Review · Reviewer_Dw3F · 2024-05-31

**Rating:** 3
**Confidence:** 3

**Summary:**

This paper works on estimating 3D interacted hand poses. Due to the observation that both appearance and position features are heterogeneous, the authors aim to decouple both features via facilitating interaction within  each feature type and those between both feature types.

**Strengths:**

- The proposed pipeline outperforms current methods on InterHand2.6M dataset and also shows strong generalization ability on in-the-wild images.

**Limitations:**

- The paper is not presented clearly in terms of method details and results presentation.
- The pipeline seems to achieve outstanding performance, but it is not easy to follow the methodology behind the model design. Based on the textual and visual materials, I understand the authors' intend to better model both appearance and position features instead of using direct fusion. However, the materials does not make it easy to follow. The authors might want to provide more concise or easy-to-follow clarification.
- What are "single", "two", "all" in results tables?
- In Section 4.3.2, does it refer to Table 2?

**Suitability:**

3

---

### Meta-Review · Area_Chair_tMkK · 2024-06-30

**Recommendation:** Accept (Poster)
**Confidence:** 3

**Metareview:**

This paper received contrasting reviews, originally they were BR, BA, WR, BA, and after rebuttal the first BR and BA remained the same, but the latter two were not finalized. In practice, this paper is on the fence.

Main remarks regard the low clarity of the method's description, the need of better explanation and justification of some parts of the approach, the revision of the experimental results, and some other aspects (e.g., explainability, computational cost).
Good aspects regard the good novelty of the approach and superior performance.

The rebuttal addressed all the comments in a good manner to the AC opinion, clarifying the raised main points quite well.
As a results, this paper can be considered acceptable for publication to the ACM MM 24 conference.